# Indoor Pedestrian Location via Factor Graph Optimization Based on Sliding Windows

**DOI:** 10.3390/s25175545

**Published:** 2025-09-05

**Authors:** Yu Cheng, Haifeng Li, Xixiang Liu, Shuai Chen, Shouzheng Zhu

**Affiliations:** 1School of Instrument Science and Engineering, Southeast University, Nanjing 210096, China; 230239008@seu.edu.cn; 2Academic Affairs Office, Army Academy of Border and Coastal Defence, Kunming Campus, Kunming 650207, China; lihaifeng517@sohu.com; 3School of Automation, Nanjing University of Science and Technology, Nanjing 210094, China; chenshuai@njust.edu.cn; 4Hangzhou Institute for Advanced Study, University of Chinese Academy of Sciences, Hangzhou 310024, China; zhushouzheng@ucas.ac.cn

**Keywords:** GNSS, PDR, FGO, transformer, TCN

## Abstract

Global navigation satellite systems (GNSS) can provide high-quality location information in outdoor environments. In indoor environments, GNSS cannot achieve accurate and stable location information due to the obstruction and attenuation of buildings together with the influence of multipath effects. Due to the rapid development of micro-electro-mechanical system (MEMS) sensors, today’s smartphones are equipped with various low-cost and small-volume MEMS sensors. Therefore, it is of great significance to study indoor pedestrian positioning technology based on smartphones. In order to provide pedestrians with high-precision and reliable location information in indoor environments, we propose a pedestrian dead reckoning (PDR) method based on Transformer+TCN (temporal convolutional network). Firstly, we use IMU (inertial measurement unit)/PDR pre-integration to suppress the inertial navigation divergence. Secondly, we propose a step length estimation algorithm based on Transformer+TCN. The Transformer and TCN networks are superimposed to improve the ability to capture complex dependencies and improve the generalization and reliability of step length estimation. Finally, we propose factor graph optimization (FGO) models based on sliding windows (SW-FGO) to provide accurate posture, which use accelerometer (ACC)/gyroscope/magnetometer (MAG) data to establish factors. We designed a fusion positioning estimation test and a comparison test on step length estimation algorithm. The results show that the fusion method based on SW-FGO proposed by us improves the positioning accuracy by 29.68% compared with the traditional FGO algorithm, and the absolute position error of the step length estimation algorithm based on Transformer+TCN in pocket mode is mitigated by 42.15% compared with the LSTM algorithm. The step length estimation model error of Transformer+TCN is 1.61%, and the step length estimation accuracy is improved by 24.41%.

## 1. Introduction

With the development of Internet of Things technology, humans have more and more been positioning their needs in indoor environments. Data show that humans spend 80% of their time in indoor environments [1]. Indoor pedestrian positioning technology refers to the use of different technologies to position and track people in various complex indoor environments [2]. GNSS can provide high-quality location information in outdoor environments. However, due to the obstruction of buildings and the influence of multipath effects, GNSS cannot achieve accurate and stable positioning. Therefore, people have an urgent need for accurate indoor positioning and seamless indoor and outdoor positioning methods. In addition, MEMS sensors have become low-cost, small in size, and low in power consumption due to the rapid development of MEMS sensors. Today’s smartphones are already equipped with various MEMS sensors, such as accelerometers, gyroscopes, barometers, cameras, microphones, ambient light sensors, magnetometers, etc. [3]. The widespread popularity of smartphones means that research on indoor pedestrian autonomous navigation methods on smartphones has important commercial and social value.

Indoor pedestrian positioning technology includes ultra-wide band (UWB), radio frequency, WiFi, inertial navigation, geomagnetism, etc., and is gradually applied to all walks of life [4]. However, with the development of society, a single indoor positioning technology cannot meet people’s requirements for higher positioning accuracy, more comprehensive coverage, and lower costs [5,6,7]. Pedestrian dead reckoning is a method that uses step length and direction of travel to calculate the relative position of pedestrians. It is widely used in the field of indoor pedestrian positioning [8,9]. It can estimate the number of steps and cadence of pedestrians by collecting sensor data from motion sensors, such as accelerometers or even from electromyography sensor to achieve gait detection. Based on this, the pedestrian’s heading is determined by data collected by gyroscopes and magnetometers. It is estimated that the location coordinates of the pedestrian can eventually be calculated. Scholars from navigation and positioning research community have conducted various research on pedestrian dead reckoning methods. Normally, their methods can be classified into two major research directions: step detection including pace/stride/gait detection and step length estimation, and heading estimation. This paper mainly improves the step length estimation.

Step length estimation methods are mainly divided into direct methods and indirect methods [10,11]. In direct methods, zero velocity updates (ZUPTs) [12] and their improved algorithms are commonly used to reduce error accumulation. However, ZUPTs require the IMU to be placed at the foot of the pedestrian, which is not suitable for practical applications. In terms of indirect methods, there are many related studies on statistical prediction methods using machine learning. Reference [13] proposed a KFBPE algorithm that can self-learn online and update the step length in real time. After combining it with the near-ultrasound positioning method, it can achieve accurate positioning within an error of less than 25 cm. Reference [14] proposed a deep learning method to estimate the stride length of healthy users and elderly patients. This method relies on inertial sensors fixed on the feet of pedestrians and estimates the stride length through convolutional neural networks. The step length results have high accuracy, but additional hardware equipment needs to be installed. Reference [15] uses smartphones to collect data and uses online sequential-extreme learning machine (OS-ELM) to estimate the step length, but ignores the correlation between acceleration sequences. Sui et al. [16] proposed a single convolutional neural network (CNN) model to predict running and walking stride length, achieving a better mean percentage error (4.78%) in running and walking stride length regression. Sun et al. [17] proposed a deep recurrent neural network (RNN)-based stride length estimation for underground personnel, achieving a relative error of 5.9%. Wang et al. [18] proposed a stride length estimation method based on a long short-term memory (LSTM) network and a denoising autoencoder (DAE). The LSTM network was used to exploit temporal dependencies and extract important feature vectors from corrupted inertial sensor observations. At an 80% confidence level, the stride length error rate was 4.59%. However, the above machine learning methods usually need to collect a large amount of training data, collect the walking step length of each step of pedestrians for accurate marking, and then train a suitable model. On the one hand, since the model needs to adapt to the different step length characteristics of different pedestrians, more experimenters are required to participate in data collection. On the other hand, the current step length acquisition method generally adopts double integration and recalibration of IMU data, or uses high-precision speed measurement equipment, which will inevitably lead to error accumulation and insufficient precision in the calculation process.

At present, since each single pedestrian positioning technology has its advantages and disadvantages, researchers try to combine multiple positioning technologies to improve the performance of pedestrian navigation in a complex environment [19]. However, multi-source fusion positioning also faces specific challenges, such as data inconsistency between sensors, sensor calibration issues, and the design and optimization of data fusion algorithms. Therefore, in practical applications, it is necessary to comprehensively consider various factors and reasonably select sensor combinations and fusion algorithms to achieve the best positioning performance. J. Coulin [20] proposed an extension of the tightly coupled multi-state constrained Kalman filter (MSCKF) to achieve long-term indoor positioning by combining visual, inertial, and magnetometer data. Li et al. [21] demonstrated the advantages of FGO in improving positioning accuracy by comparing it with the extended Kalman filter (EKF) in the PPP-B2b/INS integrated navigation application. Hu et al. [22] propose a decentralized multi-sensor information fusion framework for hypersonic vehicle navigation using a robust unscented Kalman filter (UKF), which showed excellent performance under dynamic maneuvers. Gao et al. [23] present a new methodology of distributed state fusion by using the sparse-grid quadrature filter to deal with the fusion estimation problem for multi-sensor nonlinear systems. The proposed methodology of distributed fusion can obtain higher fusion estimation accuracy in a flexible way, leading to improved fusion performance for multi-sensor nonlinear systems. Gao et al. [24] present an unscented Kalman filter (UKF) based multi-sensor optimal data fusion methodology for INS/GNSS/CNS integration based on a nonlinear system model. The proposed methodology refrains from the use of covariance upper bound to eliminate the correlation between local states. Its efficacy is verified through simulations, practical experiments and comparison analysis with the existing methods for INS/GNSS/CNS integration. Hu et al. [25] present a matrix weighted multi-sensor data fusion methodology with a two-level structure for an INS/GNSS/CNS integrated navigation system. Compared with the data fusion algorithm based on generic weighting matrices, the computational load involved in the one based on diagonal weighting matrices is significantly reduced. Gao et al. [26] present a new multi-sensor data fusion methodology for INS/GPS/SAR integrated navigation systems. This methodology combines local decentralized fusion with global optimal fusion to enhance the accuracy and reliability of integrated navigation systems.

FGO has been shown to be able to handle large-scale problems and multiple iterations in multi-sensor data fusion, making it an effective means to improve the performance of navigation systems [27]. Wang et al. [28] used the factor graph in SLAM for fusion positioning and fused the position and heading information of WIFI and PDR. The final positioning error was only 1.22 m. Zuo et al. [29] used the graph optimization method to optimize the navigation information as a whole. Compared with the filtering method, this method has lower real-time performance and larger calculation complexity, but has better positioning accuracy. Christian Merfels et al. [30] proposed a graph optimization method based on sliding windows factor for autonomous vehicles to achieve plug-and-play access to multiple sensor information. Wen et al. [31] summarized the advantages and disadvantages of factor graph optimization and Kalman filtering, and proposed an inertial/satellite tightly coupled integrated navigation method based on factor graph optimization. Tong Qin et al. [32] proposed a global state estimator based on factor graph optimization, which uses absolute measurement information and relative measurement information to achieve high-precision positioning. Ruben Mascaro et al. [33] optimized the factor graph between the relative pose obtained by inertial/visual odometry and satellite data to obtain accurate global navigation information. In order to solve the problem of observation anomalies in complex underwater environments, Xun Dong et al. [34] proposed an improved factor graph based on Mahalanobis distance anomaly detection and constructed a new INS/DVL/USBL combined factor graph model with embedded anomaly detection nodes. Zehua Li [35] proposed an improved factor graph model based on measurement rationality and used this model to combine pedestrian dead reckoning (PDR) with magnetic vector fingerprint to achieve indoor navigation. C. Pan [36] proposed a smartphone-based vision/MEMS-IMU/GNSS tightly coupled seamless positioning method to provide continuous and reliable positioning services in dynamic and complex environments through FGO. Changhui Jiang et al. [37] used the FGO algorithm to fuse PDR step size and GNSS to achieve precise positioning in order to solve the problem of heading angle error and inconsistency between smartphones and pedestrians’ walking directions. Ming Wang et al. [38] proposed a neural network-assisted factor graph optimization (NN-FGO) method for collaborative pedestrian navigation (CPN) to solve the problem of radio positioning’s heavy reliance on infrastructure, which effectively reduced the average position error.

Based on the current research pain points, we propose a lightweight model based on Transformer+TCN to achieve step length estimation. By learning the step length features in the input time-series data, the accuracy and robustness of step length estimation are improved. In addition, we factor graph fuse IMU, PDR and magnetometer information, suppress inertial navigation divergence through IMU/PDR pre-integration, PDR constrains absolute position information, and the magnetometer provides more accurate attitude information, thereby achieving high-precision positioning.

The main contributions can be summarized as follows:(1)We propose a lightweight stride length estimation algorithm based on Transformer+TCN and examine the impact of different pedestrian activity differences on stride length estimation accuracy. We train the Transformer+TCN model using IMU data and corresponding movement distance data to learn stride length characteristics from the input data time series. Experimental results show that the step length estimation model error of Transformer+TCN is 1.61%, and the step length estimation accuracy is improved by 24.41%, which can adapt to the different stride length characteristics of different pedestrians and improve the accuracy and robustness of stride length estimation.(2)We propose a sliding window-based factor graph optimization model (SW-FGO) that integrates information from IMU, PDR, and magnetometer. Experimental results show that compared with the traditional FGO model, the SW-FGO model improves positioning performance by 29.68%.(3)Since low-cost IMUs have greater zero bias and noise, we are inspired by the IMU/ODO pre-integration algorithm in vehicle-mounted navigation and positioning. In the optimization of smart handheld terminals, we obtain the mileage increment information of the PDR to constrain the IMU relative mileage. Experimental results show that the IMU/PDR pre-integration improves the positioning accuracy by 21.6% compared with the IMU pre-integration.(4)We collected data from 10 users in four different terminal postures (flat, talking, hand-free, and pocket). Experimental results validated the effectiveness of the proposed Transformer+TCN step-size estimation algorithm and sliding-window-based factor graph optimization model. The experimental results show that the proposed method can effectively improve positioning accuracy.

## 2. Theory and Methodology

### 2.1. Overall Architecture

The schematic diagram of the PDR system is shown in Figure 1. $\left(E_{0},N_{0}\right)$ is the coordinate of the starting point, $\left(E_{n},N_{n}\right)$ is the coordinate of the position after walking *n* steps, $l_{n}$ is the step length after walking *n* steps, and $\theta_{n}$ is the heading angle after walking *n* steps. The position of the $\left(E_{n},N_{n}\right)$ is(1)$$\left\{\begin{array}{c} E_{n}=E_{n-1}+l_{n}\cos\theta_{n}\\ N_{n}=N_{n-1}+l_{n}\sin\theta_{n} \end{array}\right.$$

This article mitigates the IMU error through IMU/PDR pre-integration, establishes a factor graph model of pre-integration on magnetometer, accelerometer and PDR position, optimizes the pose information, reduces positioning error, and provides zero speed and zero angular speed simultaneously. The positioning framework is shown in Figure 2. Accelerometers and magnetometers mainly provide attitude information and quasi-static field detection is added to prevent major interference with attitude in a magnetic field interference environment. When stationary, zero speed and zero angular velocity models are provided to optimize the stationary state estimation. The relative trajectory of the PDR is provided to the IMU for pre-integration to suppress the divergence caused by the low-cost IMU.

### 2.2. IMU/PDR Pre-Integration

In vehicle navigation and positioning, the IMU/odometer pre-integration algorithm directly uses the original mileage increment information measured by odometer to avoid errors caused by converting mileage increments into speed and, at the same time, constructs a pre-integration form suitable for nonlinear optimization problems. Inspired by the IMU/odometer pre-integration algorithm in vehicle navigation and positioning, in the optimization of smart handheld terminals, relative mileage constraints are imposed on the IMU by obtaining the mileage increment information of the PDR. The differential equation for PDR pre-integration is expressed as(2)$$\overset{g}{r_{b_{t}}^{b_{k-1}}}=v_{pdr}$$

$v_{pdr}$ is the speed and $\overset{g}{r_{b_{t}}^{b_{k-1}}}$ is the human mileage increment. Error analysis can be obtained as(3)$$\overset{g}{r_{b_{t}}^{b_{k-1}}}+\delta r_{b_{t}}^{b_{k-1}}=v_{pdr}+w_{v}$$

The differential equation of IMU/PDR pre-integration error is expressed as(4)$$\overset{g}{\delta x_{t}^{t_{k}}}=(I+F_{t}\Delta t)\delta x_{t}^{t_{k-1}}+G_{t}\Delta tw_{t}$$(5)$$\delta x_{t}^{t_{k}}={\left[\begin{array}{cccccc} \delta p_{b_{t}}^{b_{k}} & \delta v_{b_{t}}^{b_{k}} & \delta \theta_{b_{t}}^{b_{k}} & \delta b_{g} & \delta b_{a} & \delta r_{b_{t}}^{b_{k}} \end{array}\right]}^{T}$$(6)$$w={\left[\begin{array}{ccccc} w_{g} & w_{a} & w_{cg} & w_{ca} & w_{v} \end{array}\right]}^{T}$$(7)$$F_{t}=\left[\begin{array}{cccccc} 0_{3\times 3} & I_{3\times 3} & 0_{3\times 3} & 0_{3\times 3} & 0_{3\times 3} & 0_{3\times 3}\\ 0_{3\times 3} & 0_{3\times 3} & -R\left\{\;{\hat{q}}_{b_{i}}^{b_{k}}\right\}\left[({\hat{a}}_{i+1}-b_{a})\times \right] & 0_{3\times 3} & -R\left\{\;{\hat{q}}_{b_{i}}^{b_{k}}\right\} & 0_{3\times 3}\\ 0_{3\times 3} & 0_{3\times 3} & -\left[({\hat{\omega }}_{i+1}-b_{g})\times \right] & I_{3\times 3} & 0_{3\times 3} & 0_{3\times 3}\\ 0_{3\times 3} & 0_{3\times 3} & 0_{3\times 3} & -I_{3\times 3}/\tau_{g} & 0_{3\times 3} & 0_{3\times 3}\\ 0_{3\times 3} & 0_{3\times 3} & 0_{3\times 3} & 0_{3\times 3} & -I_{3\times 3}/\tau_{a} & 0_{3\times 3}\\ 0_{3\times 3} & 0_{3\times 3} & 0_{3\times 3} & 0_{3\times 3} & 0_{3\times 3} & 0_{3\times 3} \end{array}\right]$$(8)$$G_{t}=\left[\begin{array}{ccccc} 0_{3\times 3} & 0_{3\times 3} & 0_{3\times 3} & 0_{3\times 3} & 0_{3\times 3}\\ R\left\{\;{\hat{q}}_{b_{i}}^{b_{k}}\right\} & 0_{3\times 3} & 0_{3\times 3} & 0_{3\times 3} & 0_{3\times 3}\\ 0_{3\times 3} & I_{3\times 3} & 0_{3\times 3} & 0_{3\times 3} & 0_{3\times 3}\\ 0_{3\times 3} & 0_{3\times 3} & I_{3\times 3} & 0_{3\times 3} & 0_{3\times 3}\\ 0_{3\times 3} & 0_{3\times 3} & 0_{3\times 3} & I_{3\times 3} & 0_{3\times 3}\\ 0_{3\times 3} & 0_{3\times 3} & 0_{3\times 3} & 0_{3\times 3} & I_{3\times 3} \end{array}\right]$$

$\delta p_{b_{t}}^{b_{k}}$ is the position error. $\delta v_{b_{t}}^{b_{k}}$ is the velocity error. $\delta \theta_{b_{t}}^{b_{k}}$ is the attitude error. $\delta r_{b_{t}}^{b_{k}}$ is the PDR error. $\delta b_{g}$ is gyroscope bias error. $\delta b_{a}$ is the accelerometer bias error. $w_{v}$ is PDR noise. x is the IMU pre-integration state. w is the system noise vector. F is the system state matrix. G is the system noise matrix. Δt is the time interval. $w_{g}$ is the measurement noise of the gyroscope. $w_{a}$ is the measurement noise of the accelerometer. $w_{cg}$ is the white noise of the gyroscope bias. $w_{ca}$ is the white noise of the accelerometer bias. $w_{v}$ is the PDR noise. $R\left\{\;{\hat{q}}_{b_{i}}^{b_{k}}\right\}$ is the corresponding rotation matrix of ${\hat{q}}_{b_{i}}^{b_{k}}$. ${\hat{a}}_{i+1}$ is the acceleration data. ${\hat{\omega }}_{i+1}$ is the gyroscope data. $b_{g}$ is the gyroscope bias. $b_{a}$ is the accelerometer bias. $\tau_{g}$ and $\tau_{a}$ are the correlation times of the accelerometer and gyroscope biases, respectively. Each time the residual is calculated, the IMU’s pre-integration amount will be corrected.(9)$${\hat{p}}_{b_{k+1}}^{b_{k}}=p_{b_{k+1}}^{b_{k}}+J_{b_{g}}^{p}\delta b_{g}+J_{b_{a}}^{p}\delta b_{a}$$(10)$${\hat{v}}_{b_{k+1}}^{b_{k}}=v_{b_{k+1}}^{b_{k}}+J_{b_{g}}^{v}\delta b_{g}+J_{b_{a}}^{v}\delta b_{a}$$(11)$${\hat{q}}_{b_{k+1}}^{b_{k}}=q_{b_{k+1}}^{b_{k}}⊗\left[\begin{array}{c} 1\\ \frac{1}{2}J_{b_{g}}^{\theta }\delta b_{g} \end{array}\right]$$(12)$$\delta b_{g}=b_{gb_{k+1}}-b_{gb_{k}}$$(13)$$\delta b_{a}=b_{ab_{k+1}}-b_{ab_{k}}$$

In the formula, $J_{b_{g}}^{p}$ is the corresponding part of $J_{t_{k},t_{k-1}}$, $J_{b_{a}}^{p}$, $J_{b_{g}}^{v}$, $J_{b_{a}}^{v}$, $J_{b_{g}}^{\theta }$ have similar meanings, $b_{gb_{k+1}}$, $b_{ab_{k+1}}$ are, respectively, the zero bias of the gyroscope and the zero bias of the accelerometer used when constructing the IMU/PDR pre-integration at the moment. IMU/PDR pre-integration constrained cost function is expressed as(14)$$e\left(x_{k-1},x_{k}\right)=\left[\begin{array}{c} R\left\{\;{\hat{q}}_{b_{k}}^{b_{k-1}}\right\}\left(p_{b_{k}}^{n}n-p_{b_{k-1}}^{n}-v_{b_{k-1}}^{n}\Delta t_{k}^{k-1}+\frac{1}{2}g^{m}{\left(\Delta t_{k}^{k-1}\right)}^{2}\right)-{\hat{p}}_{b_{k}}^{b_{k-1}}\\ R\left\{\;{\hat{q}}_{b_{k}}^{b_{k-1}}\right\}\left(v_{b_{k}}^{n}-v_{b_{k-1}}^{n}+g^{n}\Delta t_{k}^{k-1}\right)-{\hat{v}}_{b_{k}}^{b_{k-1}}\\ 2{\left[\left({\left(q_{b_{k-1}}^{n}\right)}^{-1}⊗q_{b_{k}}^{n}\right)⊗{\left({\hat{q}}_{b_{k}}^{b_{k-1}}\right)}^{-1}\right]}_{xyz}\\ b_{gb_{k}}-b_{gb_{k-1}}+b_{gb_{k-1}}/\tau_{g}\\ b_{ab_{k}}-b_{ab_{k-1}}+b_{ab_{k-1}}/\tau_{a}\\ R\left\{\;{\hat{q}}_{b_{k}}^{b_{k-1}}\right\}\left(p_{b_{k}}^{n}-p_{b_{k-1}}^{n}\right)-R\left\{\;{\hat{q}}_{b_{k}}^{b_{k-1}}\right\}{\hat{r}}_{b_{k}}^{b_{k-1}} \end{array}\right]$$

### 2.3. Step Length Estimation Algorithm Based on Transformer+TCN

Step length estimation is a complex nonlinear relationship that is difficult to describe using mathematical formulas accurately. Neural networks are very suitable for handling nonlinear problems, so we propose a step estimation algorithm based on Transformer+TCN neural networks as shown in Figure 3. In the offline training phase, the intelligent handheld terminal sensor data and the plane coordinates collected by the UWB module are synchronously collected. The accelerometer and gyroscope data measured by the IMU are used as training data, and the distance calculated by the UWB is used as training labels to train the step size model. In the prediction stage, by inputting IMU data from the intelligent handheld terminal and outputting the predicted value through the step size estimation model, a more accurate pedestrian step size is obtained.

We have established a step size estimation model for the Transformer+TCN neural network. The transformer filters out noise from the accelerometer and gyroscope data and learns useful features for step size estimation. Then, the obtained data are combined with the northeast sky velocity and quaternion calculated through inertial navigation and further learned step size features through a TCN network. In this paper, two independent TCN networks are used for processing. One maps the denoised IMU data sequence to the temporal feature space, while the other maps the inertial navigation solved data to the temporal feature space. Finally, the output feature vectors are concatenated to obtain the merged feature vectors, which are passed through the dropout layer and then through three fully connected layers to obtain the prediction step size.

#### 2.3.1. Feature Extraction Based on Transformer Architecture

The Transformer model is a deep learning model based on an attention mechanism, which captures the global dependency relationship between input and output through a self-attention mechanism and can be computed in parallel to accelerate the training speed. Therefore, it is widely used in NLP and other sequence-to-sequence tasks. The Transformer model mainly consists of two parts: the encoder and the decoder. The encoder part is composed of multiple identical layers stacked together, each layer containing multi-head attention, Add and Norm, and position-wise feed forward network. Assuming the input is $x=\{x^{1},x^{2},x^{3},\cdot \cdot \cdot ,x^{N}\}$, N representing the number of input samples, the encoding process is as follows:(15)$$a^{n}=f_{\theta }(x^{n})=s(W^{\left(1\right)}x^{n}+b^{\left(1\right)})$$

Among them, $\theta =\{W^{\left(1\right)},b^{\left(1\right)}\}$ represents the weights and biases of the encoding network, $a^{n}$ represents the feature mapping obtained by the hidden layer, and s() represents the activation function. The core module of the encoder is multi-head attention. The multi-head attention mechanism is developed on the basis of the self-attention mechanism. Self-attention allows models to consider the information of all elements in the entire sequence when processing each element of the sequence. It assigns different weights to an element by calculating its correlation with other elements in the sequence and then generates weighted outputs based on these weights. The self-attention mechanism typically consists of three core components: Query (Q), Key (K), and Value (V). The attention score is obtained by calculating the similarity between the query and all keys, and these scores are then used as weighted values to generate the final output. The mathematical expression for attention is as follows:(16)$$Attention(Q,K,V)=soft\max(\frac{QK^{T}}{\sqrt{d_{k}}})\times V$$

Among them, Q, K, V is the input feature vector group and is the dimension. The multi-head attention mechanism enables the model to capture information from multiple dimensions simultaneously, enhancing its understanding of input data. The attention mechanism helps models better understand the complex relationships between elements in a sequence and improve their ability to capture complex dependencies. The parallel computing of multi-head attention improves the training and inference efficiency of the model while maintaining its adaptability to different types of data, with good flexibility and high efficiency. It is mathematically expressed as follows:(17)$$MultiHead(Q,K,V)=Concat(H_{1},...,H_{n_{h}})\times W^{o}$$

The calculation formula for each head is(18)$$H_{i}=Attention(QW_{i}^{Q},KW_{i}^{K},VW_{i}^{V})$$

In the equation, $W_{i}^{Q}\in R^{d_{m}\times d_{q}}$, $W_{i}^{K}\in R^{d_{m}\times d_{k}}$, $W_{i}^{V}\in R^{d_{m}\times d_{v}}$, $W_{i}^{O}\in R^{d_{m}\times d_{o}}$ are parameter matrices. In this experiment, we set $n_{h}=8$, $d_{m}=256$, $d_{k}=d_{v}=32$.

In the decoder part, we replace the decoder part in the traditional Transformer model with a fully connected layer to prevent the decoder layer from becoming overfitting and accumulating errors and also reduce the complexity of training.

#### 2.3.2. TCN Module

Temporal convolutional network (TCN) is a deep learning model used for processing temporal data. It is based on the idea of CNN, which extracts and learns features from time-series data through convolution operations and has achieved success in a series of time-series prediction and classification tasks. To capture long-term dependencies, TCN expands the receptive field of the convolution kernel through dilated convolution. Expansion convolution expands the receptive field of convolution by inserting “gaps” between the elements of the kernel while keeping the kernel size constant. The formula is expressed as follows:(19)$$f_{k\_d}=(d-1)\times (f_{k}-1)+f_{k}$$

Among them, $f_{k}$ is the convolution kernel size of the current layer and d is the dilation number. TCN uses causal convolution to ensure that the output of each time step depends only on its previous time step and not on the future. Assuming that the current input is $f_{k}$, and the previous input sequence is $x_{1},x_{2},\cdot \cdot \cdot ,x_{t-1}$, the convolution process can be expressed as the Formula (20).(20)$$p(x)=\prod_{t=1}^{T}p(x_{t}|x_{1},x_{2},\cdot \cdot \cdot ,x_{t-1})$$

Among them, p(x) represents the convolution function, T is the total time of the time series, and t is the time of the current time series. In addition, TCN adds residual connections to sum the input and processed results, connecting the output of the temporal convolutional network with the input to preserve more original information. This helps to solve the problem of gradient vanishing caused by an increase in network layers.

### 2.4. PDR/IMU/Magnetometer Fusion

FGO is a typical time-series model. This study optimizes it using a sliding window algorithm. The size of the sliding window boundary is called the window length, and the amplitude of the boundary change is called the sliding step size. FGO constructs a graphical model of the relationship between the system state and the observed quantity over a certain period of time, and then achieves multi-source information fusion based on posterior estimation. The maximum a posteriori estimate of the factor graph is(21)$$\underset{X}{\hat{X}=argmax}\left(p(X)\right)$$
where X is the state quantity, p(X) is the joint probability distribution function of the state quantity, and $\hat{X}$ is the maximum a posteriori probability estimate. Taking each state and quantity measurement as a factor node, the factor graph model is established as follows:(22)$$p(X)=\prod_{i}f_{i}(x_{i})$$

Considering the nonlinear least squares problem, we get(23)$$\underset{X}{argmin}f(X)≜\underset{X}{argmin}\sum_{i}{\left\|h_{i}\left(X_{i}\right)-z_{i}\right\|}_{D_{i}}^{2}$$
where f(X) is the cost function, $h_{i}\left(X\right)$ is the measurement equation, and $z_{i}$ is the measurement value. Linearize the measurement equation and get(24)$$h_{i}\left(X_{i}\right)=h_{i}\left(X_{i}^{0}+\Delta_{i}\right)\approx h_{i}\left(X_{i}^{0}\right)+H_{i}\Delta_{i}$$

Taylor expansion obtains the linear least squares problem about the state update vector and the solution of the local linear problem is obtained as follows:(25)$$\Delta^{*}≜\underset{\Delta }{argmin}\sum_{i}{\left\|h_{i}\left(X_{i}^{0}\right)+H_{i}\Delta_{i}-z_{i}\right\|}_{D_{i}}^{2}\;=\underset{\Delta }{argmin}\sum_{i}{\left\|H_{i}\Delta_{i}-\left(z_{i}-h_{i}\left(X_{i}^{0}\right)\right)\right\|}_{D_{i}}^{2}$$

Eliminating the covariance matrix $\Sigma_{i}$, we get the following formula:(26)$${\left\|e_{i}\right\|}_{D_{i}}^{2}\;≜e_{i}^{T}\Sigma_{i}^{-1}e_{i}={\left(\Sigma_{i}^{-1/2}e_{i}\right)}^{T}\left(\Sigma_{i}^{-1/2}e_{i}\right)={\left\|\Sigma_{i}^{-1/2}e_{i}\right\|}_{2}^{2}={\left\|A_{i}\Delta_{i}-b_{i}\right\|}_{2}^{2}$$(27)$$A_{i}≜\Sigma_{i}^{-1/2}H_{i}$$(28)$$b_{i}≜\Sigma_{i}^{-1/2}\left(z_{i}-h_{i}\left(X_{i}^{0}\right)\right)$$

$A_{i}$ and $b_{i}$ are the whitened Jacobian matrix and prediction error. Finally, the following standard least squares problem is obtained:(29)$$\Delta^{*}\underset{\Delta }{\;=argmin}\sum_{i}{\left\|A_{i}\Delta_{i}-b_{i}\right\|}_{2}^{2}\;=\underset{\Delta }{argmin}\|A\Delta -b{\|}_{2}^{2}$$

To solve nonlinear least squares problems, this paper uses the Levenberg–Marquardt (LM) method for optimization. The LM method allows for multiple iterations to converge within the quadratic approximation region determined by the Gauss–Newton method. It is a trust region method and can be considered a combination of the Gauss–Newton method and gradient descent. By adding a non-negative constant λ to the diagonal, we obtain(30)$$\left(A^{T}A+\lambda I\right)\Delta_{lb}=A^{T}b$$

Using scaling of the diagonal terms to provide faster convergence, we obtain(31)$$\left(A^{T}A+\lambda \mathrm{diag}\left(A^{T}A\right)\right)\Delta_{lm}=A^{T}b$$

In this paper, we establish factor graph models (SW-FGO) for multiple sensors, construct measurement models for each sensor, represent them as measurement factors, and integrate each factor into the factor graph model for pose estimation. As shown in Figure 4, IMU/PDR pre-integration is performed between each PDR factor, and magnetometer and accelerometer factors are also added at this time. If a zero-speed moment is detected, a zero-speed factor is added for optimization. This article designs sliding windows for overall optimization. Factors outside the sliding windows are transformed into prior information for the next optimization through marginalization. This article achieves high-precision pose estimation by designing a factor graph optimization algorithm to fuse sensor data.

#### 2.4.1. PDR Factor

The PDR algorithm obtains the position increment of pedestrians in a step cycle. By accumulating the position increment, the PDR position can be obtained, which is used as the observation value and IMU pre-integration to obtain the position error. The cost function is(32)$$e\left(x_{k-1},x_{k}\right)=p_{pdr}-p_{b_{k}}^{n}$$

In the formula, $p_{pdr}$ is the position obtained by the PDR algorithm, $p_{b_{k}}^{n}$ is the position obtained through pre-integration; that is, the displacement is added to the optimized position in the previous moment to obtain the current position, which is updated by position.

#### 2.4.2. Attitude Factor

In order to obtain more accurate attitude information, this article establishes factors for the data of accelerometers, gyroscopes, and magnetometers. Compared to the Euler angle method, this article adopts the quaternion method to represent the attitude, improving the update rate and preventing singular value problems. The calibrated output of the magnetometer and accelerometer are used as observation vectors to optimize the state vector. By establishing an observation model, the relationship between sensor output and state vector is established, and optimization algorithms are used to minimize observation residuals, thereby obtaining more accurate attitude estimation results. The state equation of the system is(33)$$x_{n}=F_{n|n-1}x_{n-1}+v_{n-1}$$

Among them $x_{n}$ is the system state vector, where $x_{n}={\left[\begin{array}{cccc} q_{0} & q_{1} & q_{2} & q_{3} \end{array}\right]}^{T}$, $v_{n-1}$ is the system state noise, and $E[v_{n}v_{n}^{T}]=W_{n}$. The updated equation for attitude quaternions is(34)$$F_{n/n-1}=I+\frac{1}{2}\Delta tM^{\prime }\left(w_{nb}^{n}\right)$$

In the formula, $M^{\prime }\left(w_{nb}^{n}\right)$ is(35)$$M^{\prime }\left(w_{nb}^{n}\right)=\left[\begin{array}{cccc} 0 & -\omega_{x} & -\omega_{y} & -\omega_{z}\\ \omega_{x} & 0 & \omega_{z} & -\omega_{y}\\ \omega_{y} & -\omega_{z} & 0 & \omega_{x}\\ \omega_{z} & \omega_{y} & -\omega_{x} & 0 \end{array}\right]$$

In the geographic coordinate system, the gravity field vector and the geomagnetic field vector are, respectively, represented as(36)$$a^{b}={\left[\begin{array}{ccc} 0 & 0 & g \end{array}\right]}^{T}$$(37)$$m^{n}={\left[\begin{array}{ccc} m_{nx} & m_{ny} & m_{nz} \end{array}\right]}^{T}$$

In Equation (36), g is the gravitational acceleration, $m_{nx}$, $m_{ny}$, and $m_{nz}$ are the measured values of the magnetometer in the northeast celestial coordinate system. The gravity field vector in the carrier coordinate system is(38)$$a^{b}=C_{n}^{b}a^{n}=\left[\begin{array}{c} 2g\left(q_{1}q_{3}-q_{0}q_{2}\right)\\ 2g\left(q_{2}q_{3}+q_{0}q_{1}\right)\\ g\left(q_{0}^{2}-q_{1}^{2}-q_{2}^{2}+q_{3}^{2}\right) \end{array}\right]=\left[\begin{array}{cccc} -q_{2}g & q_{3}g & -q_{0}g & q_{1}g\\ q_{1}g & q_{0}g & q_{3}g & q_{2}g\\ q_{0}g & -q_{1}g & -q_{2}g & q_{3}g \end{array}\right]\left[\begin{array}{c} q_{0}\\ q_{1}\\ q_{2}\\ q_{3} \end{array}\right]=h_{a}(q)$$

The geomagnetic field vector in the carrier coordinate system is(39)$$\begin{array}{ll} m^{b}= & C_{n}^{b}m^{n}=\left[\begin{array}{c} m_{nx}\left(q_{0}^{2}+q_{1}^{2}-q_{2}^{2}-q_{3}^{2}\right)+2m_{ny}\left(q_{1}q_{2}+q_{0}q_{3}\right)+2m_{nz}\left(q_{1}q_{3}-q_{0}q_{2}\right)\\ 2m_{nx}\left(q_{1}q_{2}-q_{0}q_{3}\right)+m_{ny}\left(q_{0}^{2}-q_{1}^{2}+q_{2}^{2}-q_{3}^{2}\right)+2m_{nz}\left(q_{2}q_{3}+q_{0}q_{1}\right)\\ 2m_{nx}\left(q_{1}q_{3}+q_{0}q_{2}\right)+2m_{ny}\left(q_{2}q_{3}-q_{0}q_{1}\right)+m_{nz}\left(q_{0}^{2}-q_{1}^{2}-q_{2}^{2}+q_{3}^{2}\right) \end{array}\right]\;\\ & =\left[\begin{array}{cccc} m_{nx}q_{0}+m_{ny}q_{3}-m_{nz}q_{2} & m_{nx}q_{1}+m_{ny}q_{2}+m_{nz}q_{3} & -m_{nx}q_{2}+m_{ny}q_{1}-m_{nz}q_{0} & -m_{nx}q_{3}+m_{ny}q_{0}+m_{nz}q_{1}\\ -m_{nx}q_{3}+m_{ny}q_{0}+m_{nz}q_{1} & m_{nx}q_{2}-m_{ny}q_{1}+m_{nz}q_{0} & m_{nx}q_{1}+m_{ny}q_{2}+m_{nz}q_{3} & -m_{nx}q_{0}-m_{ny}q_{3}+m_{nz}q_{2}\\ m_{nx}q_{2}-m_{ny}q_{1}+m_{nz}q_{0} & m_{nx}q_{3}-m_{ny}q_{0}-m_{nz}q_{1} & m_{nx}q_{0}+m_{ny}q_{3}-m_{nz}q_{2} & m_{nx}q_{1}+m_{ny}q_{2}+m_{nz}q_{3} \end{array}\right]\left[\begin{array}{c} q_{0}\\ q_{1}\\ q_{2}\\ q_{3} \end{array}\right]\\ & =h_{m}(q) \end{array}$$

Establish a cost function based on the difference between $a^{b}$, $m^{b}$ and the measured values of accelerometers and magnetometers.(40)$$e\left(x_{k-1},x_{k}\right)=\left[\begin{array}{c} {\left[a^{b}-h_{a}\left({\hat{q}}_{b_{k}}\right)\right]}^{T}\\ {\left[m^{b}-h_{m}\left({\hat{q}}_{b_{k}}\right)\right]}^{T} \end{array}\right]$$

In complex scenes, there may be magnetic field interference, which seriously affects the measured magnetic field strength. At this time, the estimated heading angle of the magnetic field is not accurate, and when there is no magnetic field interference in the scene, it is considered a quasi-static field. In order to detect the quasi-static field, the variance of the output value of the three-axis magnetometer is used for detection, and the heading error of the gyroscope is combined for judgment. The judgment conditions are(41)$$\left\{\begin{array}{c} {var}_{k:k+L}\left(m_{x}\right)+{var}_{k:k+L}\left(m_{y}\right)+{var}_{k:k+L}\left(m_{z}\right),{var}_{thod}\\ \left|\sin\left(∆\varphi_{m,k}\right)-sin(∆\varphi_{g,k})\right|sin(∆\varphi_{thod}) \end{array}\right.$$

In the formula, ${var}_{k:k+L}(m_{x})$, ${var}_{k:k+L}(m_{y})$, and ${var}_{k:k+L}(m_{z})$ are the variances of the output values of the three-axis magnetometer within the windows; the sliding windows are set to *L*; ${var}_{thod}$ is the empirical value of the variance; $\Delta \varphi_{m,k}$ is the heading angle change output by the magnetometer; $\Delta \varphi_{g,k}$ is the heading angle change calculated by the gyroscope; and $\Delta \varphi_{thod}$ is the empirical value of the heading angle difference. The empirical value can be measured without magnetic field interference. In order to prevent the deviation of the heading angle from jumping, the sine value is taken. If the magnetic field is judged as a quasi-static field, it is considered that a magnetometer is available, and a magnetometer is used for fusion. If it is not a quasi-static field, it means that the magnetometer is affected by magnetic interference, and the magnetometer will no longer be fused.

#### 2.4.3. Zero Speed Factor Model

When the carrier is stationary, the speed and angular velocity of the intelligent handheld terminal are zero, and this constraint is updated as an observation value. The activity recognition can complete the judgment of zero-speed moment mentioned earlier. The measurement equation at zero speed is(42)$$v_{b_{k}}^{n}-0=\delta v_{b_{k}}$$

In the formula, $v_{b_{k}}^{n}$ is the IMU speed and $\delta v_{b_{k}}$ is the speed error. The update of zero angular velocity is quite complex, and the relationship between attitude angle and angular velocity under the load system is(43)$$\left[\begin{array}{c} \dot{\gamma }\\ \dot{\theta }\\ \dot{\psi } \end{array}\right]=\left[\begin{array}{ccc} \cos\theta & 0 & \sin\theta \\ \sin\theta \tan\gamma & 1 & -\cos\theta \tan\gamma \\ -\sin\theta /\cos\gamma & 0 & \cos\theta /\cos\gamma \end{array}\right]\left[\begin{array}{c} \omega_{nbx}^{b}\\ \omega_{nby}^{b}\\ \omega_{nbz}^{b} \end{array}\right]$$

In the formula, γ is the pitch angle, θ is the roll angle, and ψ is the heading angle. Extract the expression for the heading angle.(44)$$\delta \dot{\psi }=\left[\begin{array}{ccc} -\frac{\sin\theta }{\cos\gamma } & 0 & \frac{\cos\theta }{\cos\gamma } \end{array}\right]\delta b_{g}$$

Points obtained during the update time are(45)$$\psi_{t+\Delta T}-\psi_{t}=\delta \psi =\left[-\frac{\sin\theta }{\cos\gamma }\Delta T\;0\;\frac{\cos\theta }{\cos\gamma }\Delta T\right]\delta b_{g}$$

The cost functions for zero velocity and zero angular velocity are(46)$$e\left(x_{k-1},x_{k}\right)=[\begin{array}{c} v_{b_{k}}^{n}\\ \left[-\frac{\sin\theta }{\cos\gamma }\Delta T\;0\;\frac{\cos\theta }{\cos\gamma }\Delta T\right]\delta b_{g} \end{array}]$$

#### 2.4.4. Sliding Windows Marginalization

In graph optimization, the number of factor graph nodes will gradually increase over time, leading to an increase in the complexity of optimization calculations. In order to control the number of optimization variables and reduce computational complexity, we use sliding windows techniques to limit the number of historical variables to be optimized. The sliding windows optimization algorithm sets a fixed-size windows and selects a portion of the latest variables for optimization during each iteration while removing outdated historical variables. Marginalization is a technique that removes variables from the optimization process while retaining information related to these variables. Specifically, when discarding variables, marginalize them into factor graphs to form marginalization constraints. These marginalization constraints can be seen as prior knowledge of the discarded variables, playing a constraining role in the optimization process. This article adopts the sliding windows method to limit the computational complexity independent of the growth of optimization variables. By marginalization processing, edge constraints are added while discarding variables to reduce information loss and retain information related to discarded variables for optimization purposes. This method maintains the accuracy and effectiveness of attitude estimation while controlling computational complexity.

The cost function *f* is a nonlinear function that can be solved iteratively. The formula is as follows:(47)ΛΔχ = g

In the formula, $\Lambda =A^{T}A$ is the measurement Jacobian matrix and $g=A^{T}b$ is the prediction error matrix.

$\chi_{a}$ is the variable that needs to be marginalized, and $\chi_{b}$ is the variable that needs to be retained.(48)$$\left[\begin{array}{cc} \Lambda_{a} & \Lambda_{b}\\ \Lambda_{b}^{T} & \Lambda_{c} \end{array}\right]\left[\begin{array}{c} \Delta \chi_{a}\\ \Delta \chi_{b} \end{array}\right]=\left[\begin{array}{c} g_{a}\\ g_{b} \end{array}\right]$$

Using Sher’s elimination, we get the following formula:(49)$$\left[\begin{array}{cc} \Lambda_{a} & \Lambda_{b}\\ 0 & \Lambda_{d} \end{array}\right]\left[\begin{array}{c} \Delta \chi_{a}\\ \Delta \chi_{b} \end{array}\right]=\left[\begin{array}{c} g_{a}\\ g_{b}-\Lambda_{b}^{T}\Lambda_{a}^{-1}g_{a} \end{array}\right]$$

The cost function for marginalization is(50)$$e\left(\chi_{b}\right)=\Lambda_{d}^{\frac{1}{2}}\left(\chi_{b}{-}^{0}\chi_{b}\right)+\Lambda_{d}^{-\frac{1}{2}}\Delta \chi_{b}$$

In the formula, $0\chi_{b}$ is the estimate of $\chi_{b}$ used for Scheer’s complement and elimination.

## 3. Implementation Details

### 3.1. Dataset

We use Huawei Mate30 Pro(Huawei Technologies Co., Ltd., Shenzhen, China) as a data acquisition device to collect accelerometer gyroscope, magnetometer, UWB data and satellite data of pedestrians while walking, and the sampling frequency is set to 50 HZ. At the same time, in order to avoid overfitting during neural network training, we collected 525 pieces of data from 10 pedestrians walking on different paths, including phone postures such as flat, talking, swinging arms, and pocket mode. We divide the collected data into training and testing datasets in a 7:3 ratio. In addition, the sensor indicators are as Table 1. The gender, height, weight and age of the 10 pedestrians are shown in the following Table 2.

Because the IMU integrated in mobile phones has low precision, the collected data contain a large amount of noise, random drift, and zero bias, so data preprocessing is required. We use a high-pass filter to estimate gravity acceleration and a low-pass filter for filtering. This effectively preserves the gait characteristics in the acceleration signal while filtering out high-frequency irrelevant noise, low-frequency zero bias, and some random drift.

### 3.2. Implementation and Training Details

In the step estimation algorithm based on the Transformer+TCN network, the training hardware platform is CPU: Intel Core i7-11800H, GPU: RTX 3060, and we use the Pytorch framework (1.12.1) to build the neural network model. For hyperparameter settings, we set the batch size to 256, the initial learning rate to 0.0003, the optimizer Adam, and a dropout strategy of 0.2, with the loss function using MSE. In order to verify the effectiveness of the step length estimation model based on transformer+TCN mentioned in Section 2, the prediction error of the Transformer+TCN method was first verified using data from 10 users. Then, the influence of different terminal postures on the proposed step length estimation method was analyzed, and the Transformer+TCN, Weinberg, and LSTM methods were compared, and the absolute position error was calculated. In order to verify the effectiveness of the data fusion method based on sliding window factor graph proposed in Section 2, we built a multi-source fusion data acquisition platform based on smart handheld terminals, selected a complex environment as the test area, and designed different levels of fusion positioning estimation experiments, including quasi-static field comparison experiments, sliding window number comparison experiments, pre-integration positioning effect experiments and positioning accuracy comparison experiments to verify the effectiveness of the data fusion positioning method based on sliding window factor graph.

## 4. Experiments

### 4.1. Experimental Study on Step Estimation Algorithm Based on Transformer+TCN

We used a Huawei Mate 30 Pro as a data acquisition device to collect gyroscope, accelerometer, magnetometer, UWB data from pedestrians while walking. We use relative error to measure the performance of our step-size estimation model:(51)$$e=\frac{\left|y_{r}-\sum_{i=1}^{N}{\hat{y}}_{i}\right|}{y_{r}}$$
where ${\hat{y}}_{i}$ is the estimated step length for the *i*-th step and $y_{r}$ is the true length.

Figure 5 shows the prediction errors for 10 users. As can be seen, the average test error for these 10 users was approximately 2.38%. Users 3, 5, 6, 7, and 8 were outliers (the red “+”), likely due to differences in their walking characteristics during test data collection compared to those in the training dataset.

In order to better evaluate the effectiveness of the proposed algorithm, a step estimation model based on traditional LSTM was trained on the same training and validation sets and compared with the step estimation model proposed in this paper on the same test data. As shown in Figure 6, the prediction errors of the traditional LSTM model and the model proposed in this paper were compared. The step estimation model based on LSTM had an error of 2.13%. In comparison, The step length estimation model error of Transformer+TCN is 1.61% and the step length estimation accuracy is improved by 24.41%. So The step length estimation model of Transformer+TCN is accurate than the traditional LSTM neural network.

As shown in Figure 7, the cumulative percentage of step error of the three models is presented, with the horizontal axis representing step error and the vertical axis representing cumulative probability. All three curves show a monotonically increasing trend, with a steep initial part indicating that the data are concentrated in a small range of values and a long tail close to 1, indicating the presence of some extreme values or outliers. A total of 90% of the errors in this model are within 0.15 m, while the errors in the other two models are both above 0.2 m. According to the experimental results, it can be seen that the model established in this article has higher accuracy and can accurately describe the true step size of pedestrians.

To validate the Transformer+TCN-based proposed PDR algorithm proposed in this article, we used UWB synchronized data acquisition as the real trajectory for comparative analysis. The walking trajectory is rectangular, with a length of 40 m and a width of 20 m. The terminal data are collected at a frequency of 50 Hz. Collect four modes: flat end, call, shake, and pocket. In order to verify the effectiveness of the algorithm proposed in this paper, the heading angle obtained by compensating the attitude sensor embedded in the intelligent handheld terminal was compared with the PDR based on Weinberg step size estimation and LSTM.

Figure 8 shows the position estimation results of the four terminal usage modes, and Figure 9 shows the cumulative percentage of position errors of the four terminal usage modes. The absolute position error statistics under four terminal usage modes are shown in Table 3. The estimated position error percentages of the Transformer+TCN-based PDR method in the four terminal modes are 3.71%, 1.28%, 1.69% and 2.80%. In contrast, the estimated error percentages of the LSTM-based PDR method are 4.22%, 1.62%, 2.93% and 4.84%, and the estimated error percentages of the Weinberg-based PDR method are 4.32%, 4.18%, 4.67% and 7.81%. The step length model based on the neural network has a better positioning effect than the nonlinear model, especially the accuracy improvement of the hand-shaking and pocket mode is more obvious, indicating that the step length predicted by the neural network is more suitable for nonlinear relationships under multiple postures. The positioning accuracy of the Transformer+TCN PDR is also improved compared with the LSTM-based PDR, indicating that the Transformer+TCN model has better performance under the same task, effectively models the dependency between time series, and has high computational efficiency.

### 4.2. Sliding Windows Test

In order to determine the size of the sliding windows, we tested the fusion positioning with sliding windows of 5, 10, and 30. We used the PDR/IMU/geomagnetic fusion positioning algorithm to obtain the trajectories with sliding windows of 5, 10, and 30, respectively, as shown in Figure 10. It can be seen that the larger the sliding windows length, the closer it fits the real trajectory.

We analyzed the impact of different numbers of sliding windows on positioning accuracy, and the results are shown in Figure 11. There is no significant difference in the positioning effect of the three different sliding windows lengths in the first 250 s. As time goes by, the larger the number of sliding windows, the smaller the position error. However, the improvement of the number of sliding windows and positioning accuracy is not linear, so it is necessary to consider the fusion time of different sliding windows comprehensively.

We compared the fusion time of different numbers of sliding windows, and the results are shown in Table 4. When the number of sliding windows is 5, the fusion time is 0.104 s. When the number of sliding windows is 10, the fusion time is 0.164 s, and when the number of sliding windows is 30, the fusion time is 0.679 s. It can be seen that the longer the sliding windows, the better the positioning effect, but the longer the time consumed for each fusion. When the number of sliding windows is 30, the timeliness of the fusion algorithm cannot be met. Considering the positioning accuracy and positioning error, we choose the number of sliding windows to be 10.

### 4.3. Positioning Accuracy Test on Fusion Algorithm

Figure 12 is the position error diagram of the EKF, AUKF, FGO and SW-FGO. In the 0–50 m range, the EKF performs better than the FGO, and in the 150–225 m range, the FGO error is smaller than the SW-FGO. However, the FGO error increases over time, with the final error of the SW-FGO reaching 9.43 m, the final error of the FGO reaching 13.41 m, the final error of the AUKF reaching 15.94 m, and the final error of the EKF reaching 17.68 m. This shows that the SW-FGO has lower position error and more accurate positioning than the EKF, AUKF, and FGO. Figure 13 shows the cumulative percentage of position errors between the SW-FGO, FGO, AUKF, EKF algorithms. A total of 70% of the errors for the SW-FGO algorithm are within 6.54 m, 70% for the FGO algorithm are within 8.11 m, 70% for the AUKF algorithm are within 11.82 m, and 70% for the EKF algorithm are within 14.28 m. This shows that the SW-FGO algorithm proposed in this paper has higher positioning accuracy and stability.

### 4.4. IMU/PDR Pre-Integration Test

The experimenter walks an L-shaped trajectory, as shown in Figure 14. Comparing the trajectory with the traditional IMU pre-integration, it is clear that the IMU/PDR pre-integration method is closer to the true trajectory. The figure also clearly shows that the IMU/PDR pre-integration method has higher positioning accuracy, indicating that the IMU/PDR pre-integration method significantly suppresses IMU errors compared to traditional IMU pre-integration, thereby improving the system’s positioning accuracy.

Next, we analyzed the position errors of IMU/PDR pre-integration and traditional IMU pre-integration. The results are shown in Figure 15. The blue error is slightly larger than the red error, 19 s after the start of walking. Subsequently, the blue error gradually becomes better than the red error. At the first turn, the errors of both methods experience a step-like jump, while the IMU/PDR pre-integration method quickly returns to normal. The final position error of the IMU pre-integration method is 4.68 m, while the IMU/PDR pre-integration method is 3.67 m, achieving a 21.6% improvement in positioning accuracy.

Finally, we compared the cumulative error percentages of the two different pre-integration methods, as shown in Figure 16. The positioning error at 80% confidence level using IMU/PDR pre-integration is 1.30 m, while the positioning error at 80% confidence level using IMU pre-integration is 2.62 m. This shows that IMU/PDR pre-integration can significantly improve positioning performance compared to traditional IMU pre-integration.

## 5. Conclusions

The motivation for the proposed deep learning-based pedestrian autonomous navigation method is that when using handheld terminals for autonomous positioning in scenarios where satellites fail, high-precision and high-reliability position information is required. We meet these requirements by proposing an optimization method of SW-FGO, which uses accelerometer/gyroscope/magnetometer data to establish factors, provides accurate pose, and adds quasi-static field detection to achieve accurate attitude estimation. In addition, we propose a step length estimation algorithm based on Transformer+TCN, which superimposes the Transformer and TCN networks to improve the ability to capture complex dependencies and improve the generalization and reliability of step length estimation.

We built a multi-source fusion data acquisition platform based on smart handheld terminals, and obtained gyroscope, accelerometer, magnetometer, UWB data and satellite data, respectively. We first verified the prediction error of the Transformer+TCN method, then analyzed the impact of different terminal postures on the proposed step length estimation method, and compared the Transformer+TCN, Weinberg, and LSTM methods. The experiment showed that the step length estimation algorithm based on Transformer+TCN reduced the step length prediction error by 24.41% compared with the LSTM method. Secondly, we selected a complex environment as the test area and designed different levels of fusion positioning estimation experiments, including sliding windows number comparison experiments, pre-integration positioning effect experiments, and positioning accuracy comparison experiments to verify the effectiveness of the data fusion positioning method based on SW-FGO. The experiments showed that the positioning error of FGO is improved by 29.68% compared with the traditional FGO method. In summary, the pedestrian autonomous navigation method based on SW-FGO provides an effective solution for the development and application of pedestrian positioning technology under satellite denial.

Future research will focus on improving positioning accuracy by integrating more sensors. For example, this can be achieved by analyzing visually captured images, extracting feature information, and inferring location. Furthermore, we will consider more complex indoor scenarios for algorithm validation and consider more diverse user hand-held device configurations.

## Figures and Tables

**Figure 1 sensors-25-05545-f001:**
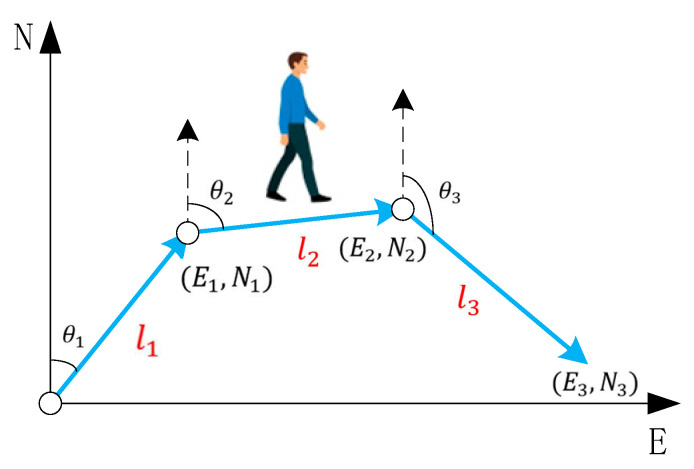
Schematic diagram of PDR system.

**Figure 2 sensors-25-05545-f002:**
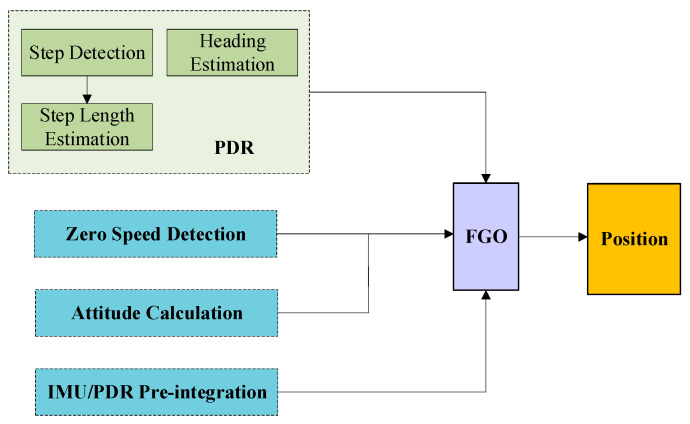
Framework diagram of pedestrian positioning algorithm based on FGO.

**Figure 3 sensors-25-05545-f003:**
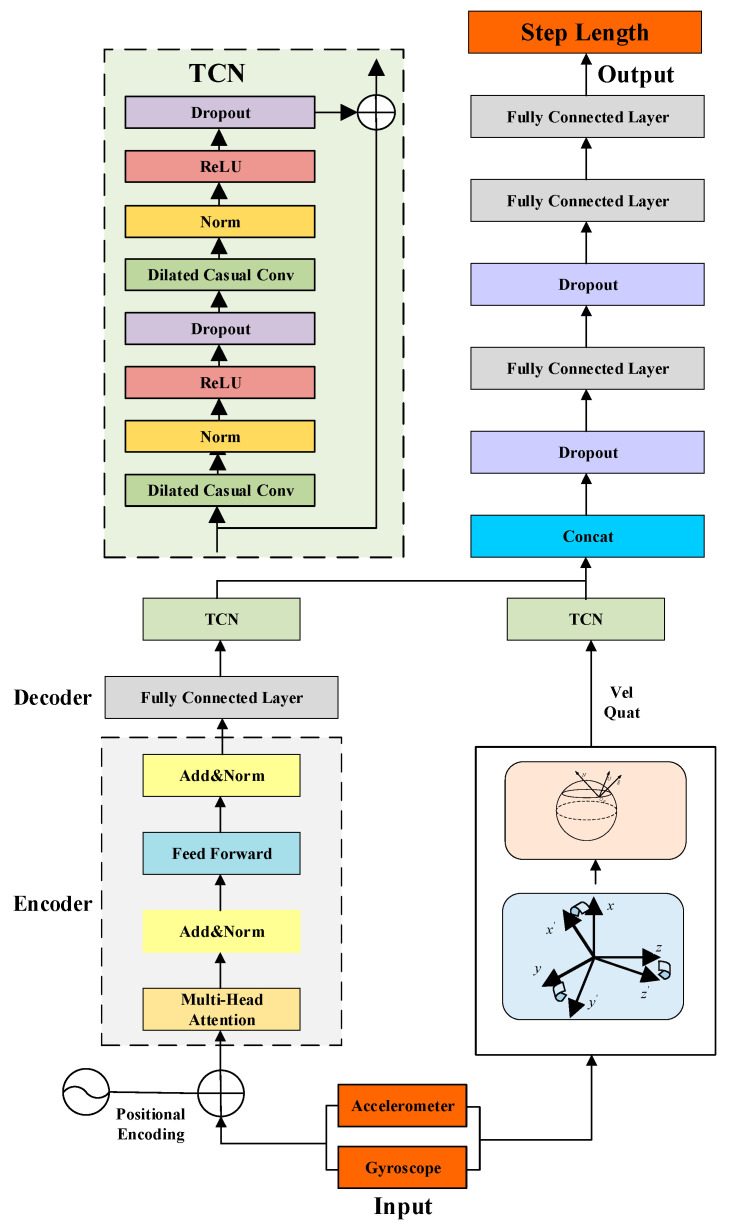
The architecture of Transformer+TCN network.

**Figure 4 sensors-25-05545-f004:**
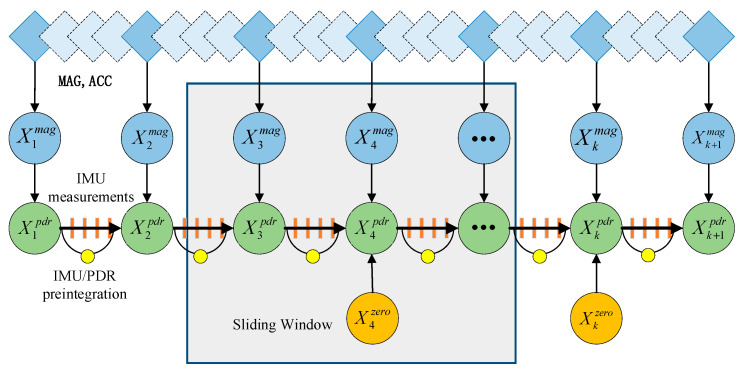
Factor graph optimization model.

**Figure 5 sensors-25-05545-f005:**
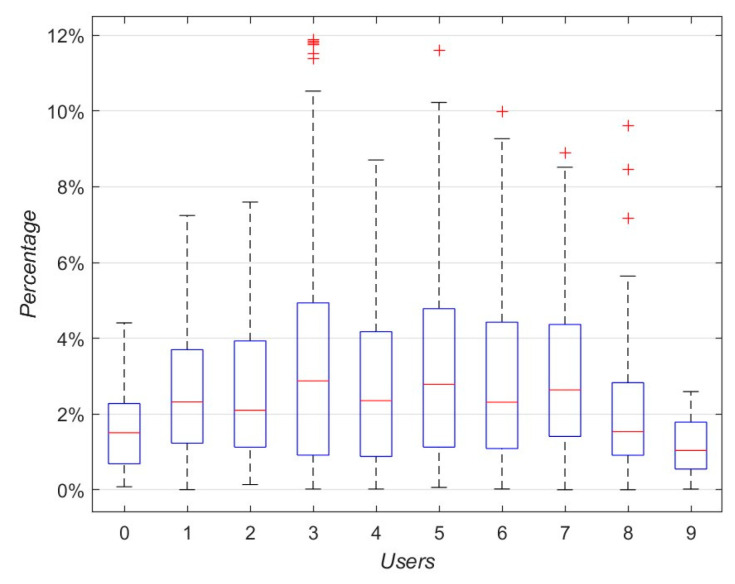
Prediction errors of different users (The red “+” is the outliers).

**Figure 6 sensors-25-05545-f006:**
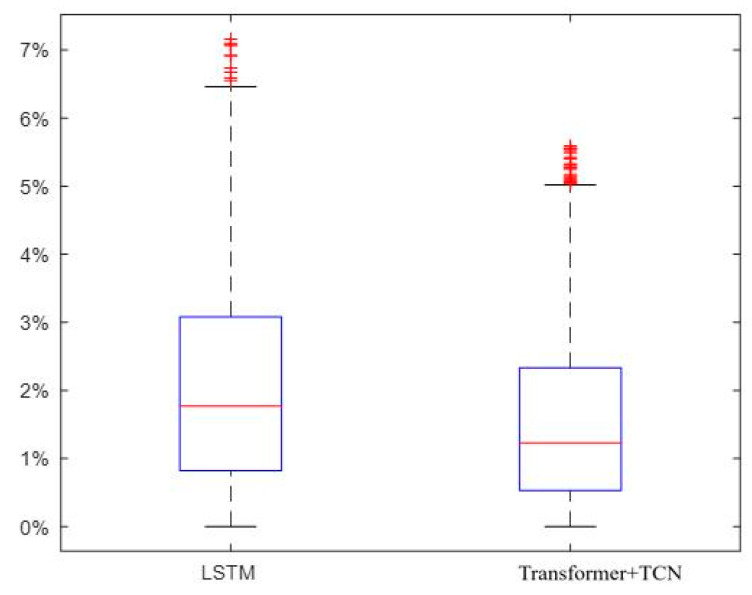
Comparison of prediction errors of different models (The red “+” is the outliers).

**Figure 7 sensors-25-05545-f007:**
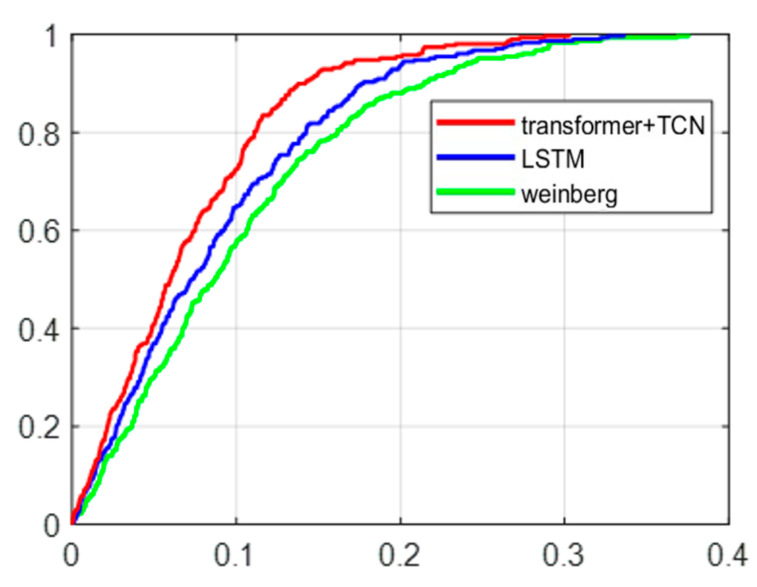
Cumulative percentage of step error of transformer+TCN, LSTM, weinberg.

**Figure 8 sensors-25-05545-f008:**
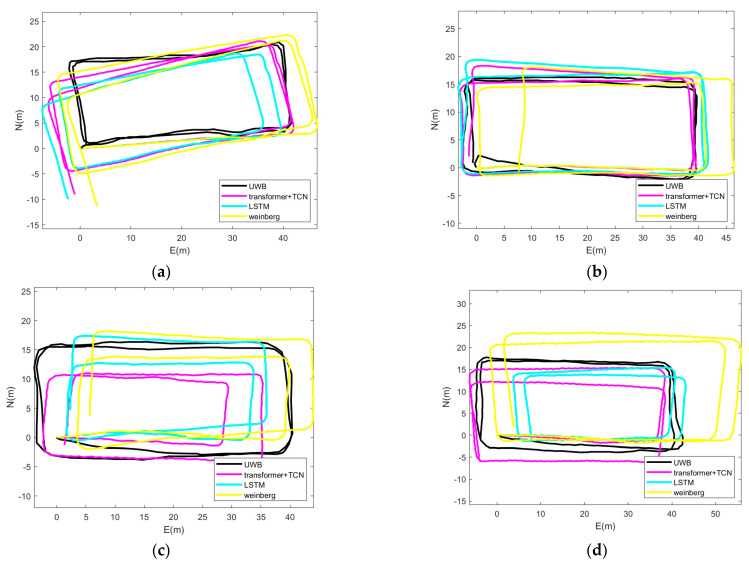
Location estimation results for four terminal usage modes. (**a**) Flat mode; (**b**) call mode; (**c**) hand-shaking mode; (**d**) pocket mode.

**Figure 9 sensors-25-05545-f009:**
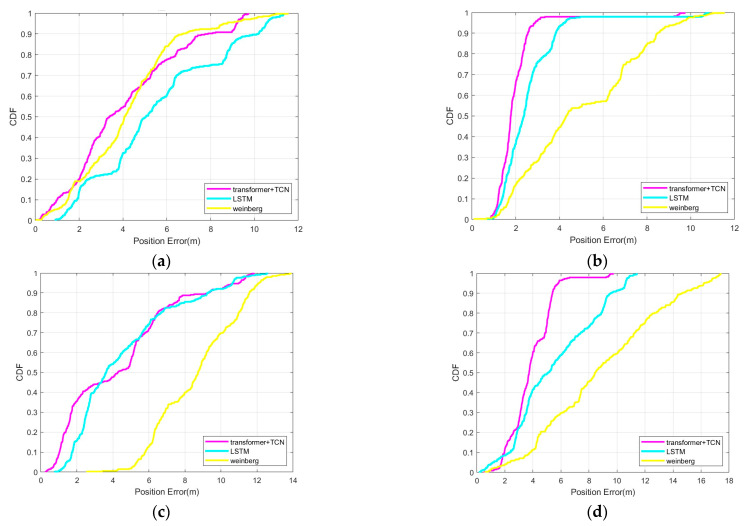
Cumulative percentage of position error of Transformer+TCN, LSTM and Weinberg. (**a**) Flat mode; (**b**) call mode; (**c**) hand-shaking mode; (**d**) pocket mode.

**Figure 10 sensors-25-05545-f010:**
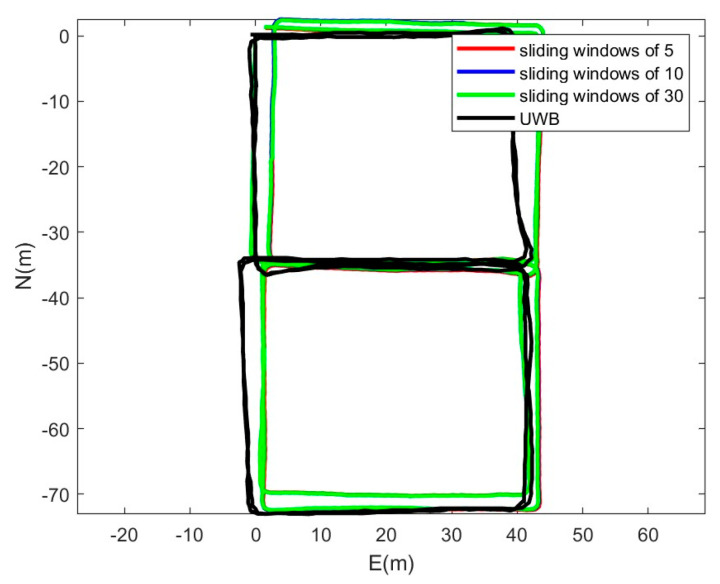
Trajectory comparison of sliding windows of 5, 10, 30.

**Figure 11 sensors-25-05545-f011:**
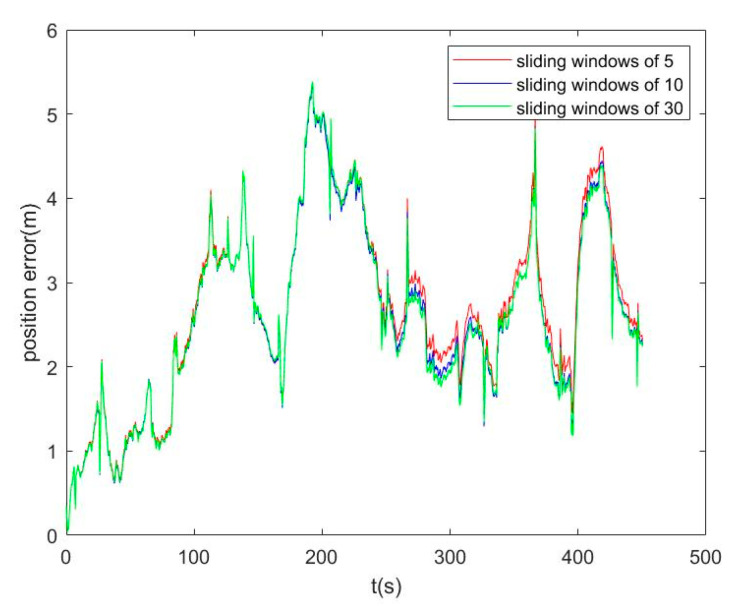
Position error comparison of sliding windows of 5, 10, 30.

**Figure 12 sensors-25-05545-f012:**
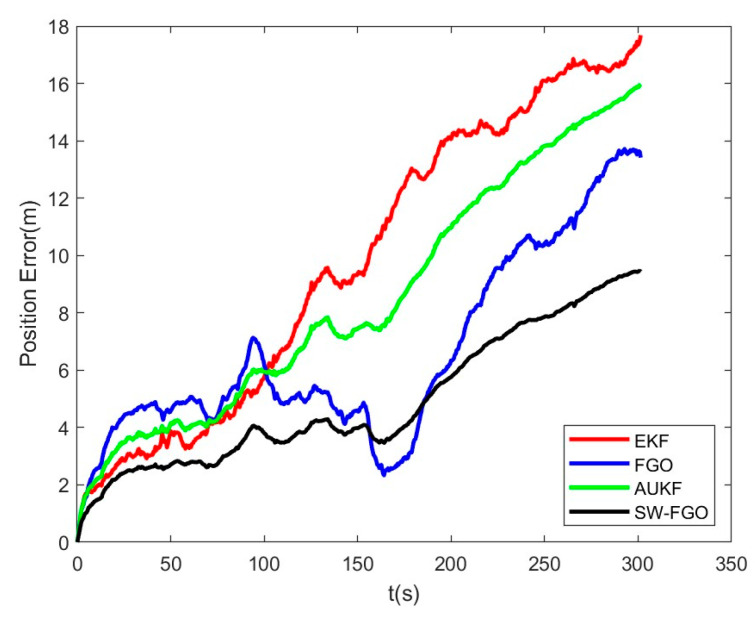
Position error of EKF, AUKF, FGO and SW-FGO.

**Figure 13 sensors-25-05545-f013:**
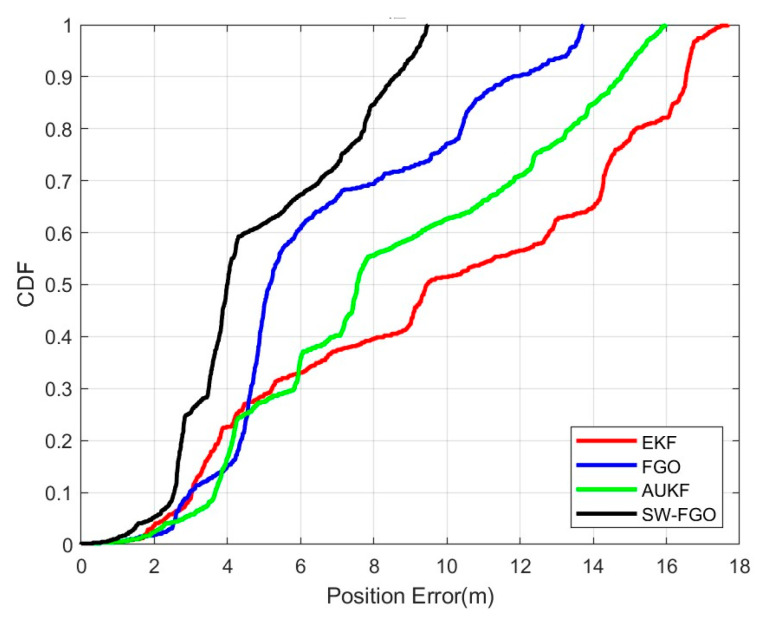
Cumulative percentage of position error of EKF, AUKF, FGO and SW-FGO.

**Figure 14 sensors-25-05545-f014:**
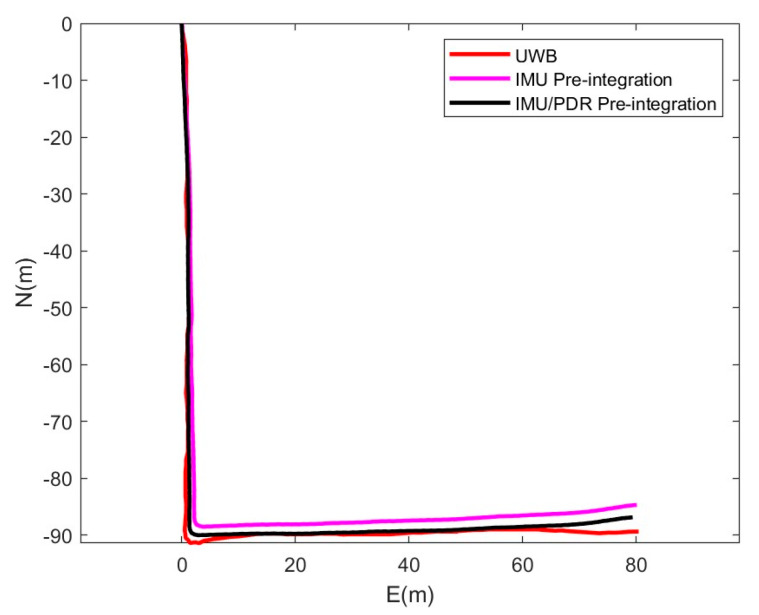
Trajectory comparison of UWB, IMU/PDR pre-integration and traditional IMU pre-integration.

**Figure 15 sensors-25-05545-f015:**
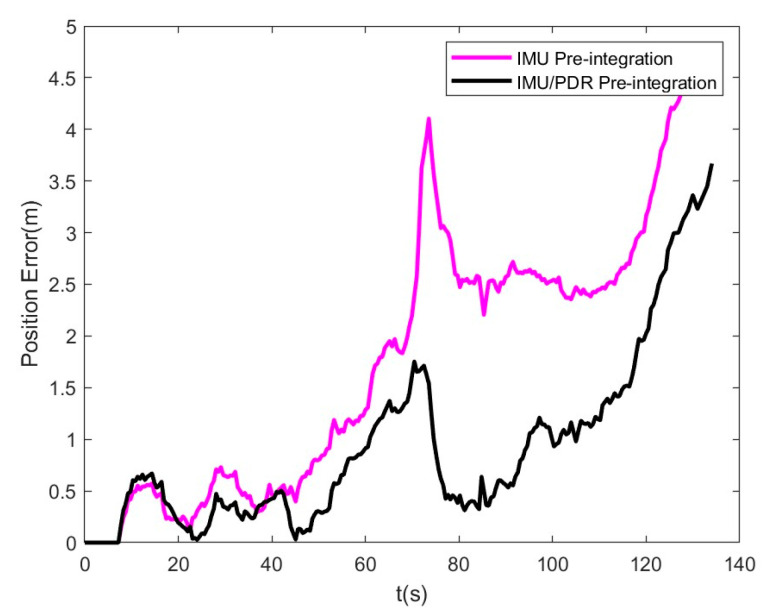
Position error of IMU/PDR pre-integration and IMU pre-integration.

**Figure 16 sensors-25-05545-f016:**
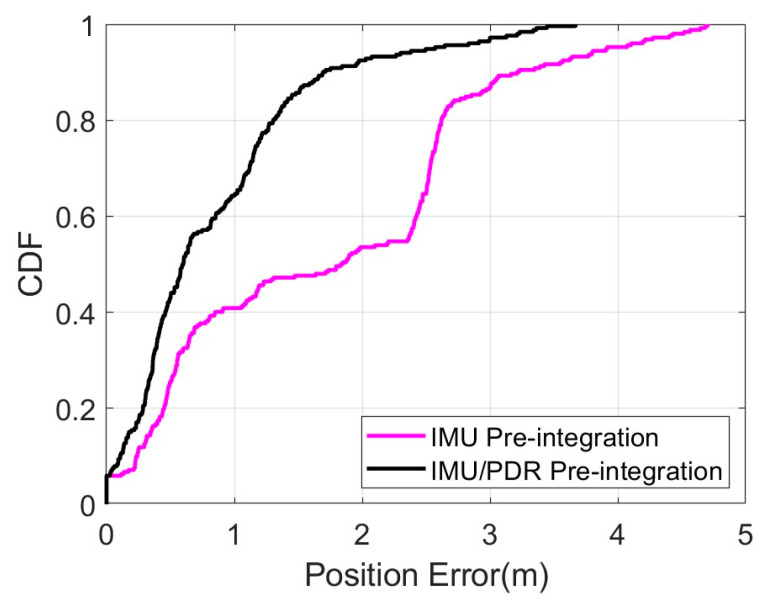
Cumulative percentage of position error of IMU/PDR pre-integration and traditional IMU pre-integration.

**Table 1 sensors-25-05545-t001:** Sensor indicators.

Sensor	Parameter	Index
Accelerometer	Range	±16 g
	Bias	25 mg
	Noise	230 μg/√Hz
Gyroscope	Range	±2000°/s
	Bias	5°/s
	Noise	0.015 dps/√Hz
Magnetometer	Range	±4800 μT
	Bias	2000 LSB
UWB	Range	200 m
	Positioning error	30 cm
GNSS	Positioning error	5 m

**Table 2 sensors-25-05545-t002:** Pedestrian list.

Number	Gender (M or F)	Height (cm)	Weight (kg)	Age
1	M	178	85	25
2	M	172	73	26
3	M	173	70	25
4	M	170	71	25
5	M	168	73	25
6	F	160	52	26
7	F	158	49	25
8	F	164	53	25
9	F	162	51	26
10	F	161	50	25

**Table 3 sensors-25-05545-t003:** Absolute position error statistics in four terminal usage modes (percentage).

Modes	Weinberg	LSTM	Transformer+TCN
flat mode	4.32%	4.22%	3.71%
call mode	4.18%	1.62%	1.28%
hand-shaking mode	4.67%	2.93%	1.69%
pocket mode	7.81%	4.84%	2.80%

**Table 4 sensors-25-05545-t004:** Single fusion time.

Sliding Windows	5	10	30
Fusion Time (s)	0.104	0.164	0.679

## Data Availability

The original contributions presented in this study are included in the article. Further inquiries can be directed to the corresponding author.

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
