# Peer review of "Indoor Pedestrian Location via Factor Graph Optimization Based on Sliding Windows"

_sensors, 2025, doi:10.3390/s25175545_

Round 1
Reviewer 1 Report
Comments and Suggestions for Authors
This paper presents a method of factor graph fusion based on sliding window for indoor pedestrian positioning. It involves the following issues:
1. The technique of factor graph fusion based on sliding window reported in this paper is not new. The state-of-the art factor graphs with the robustness against abnormal measurements have already been studied in the literature – e.g., Windowing-based factor graph optimization with anomaly detection using Mahalanobis distance for underwater INS/DVL/USBL integration, and A robust factor graph framework for navigation on PDR/magnetic field integration. Although these robust factor graphs (the first one also involves sliding window) focus on vehicle navigation, they are still applicable to the application of pedestrian positioning. However, the paper lacks the introduction of this important technique, making the literature incomplete.
2. Factor graph involves an optimization process. However, there is no discussion on how to achieve the optimization within a sliding window. There is even no discussion on the optimization involved in the traditional factor graph technique.
3. The proposed method involves a process of data training, which is done before the real-time fusion. However, in some actual scenarios with highly dynamic variations, it is difficult or impossible to conduct data training given the highly dynamic environment. How to address this issue?
4. It is quite common that the measurements of a dynamic system involve errors, outliers and even faults. How to deal with this case? Please refer to the above state-of-the-art factor graph based on sliding window and with robustness against measurement outliers and faults.
5. Factor graph is a technique in the broad field of multi-sensor fusion. However, advanced multi-sensor fusion techniques are not introduced missing - e.g., advanced fusion techniques based on random weighting estimation, robust or adaptive UKF, and sparse-grid quadrature filter. It is recommended that the authors introduce advanced fusion techniques, based on which to justify the choice of factor graph in this paper.
6. The comparison analysis is conducted with the traditional EKF. However, as advanced fusion techniques such as those mentioned above were already reported in the literature, it would be more convincing to conduct the comparison analysis them to demonstrate the improved performances. Further, the comparison analysis should be conducted with the state-of-the-art factor graph which is also based on sliding window, but with robustness against abnormal measurements.
7. No timing performance is analysed. Is the proposed method able to achieve real-time performance?
Reviewer 2 Report
Comments and Suggestions for Authors
The manuscript is devoted to indoor pedestrian navigation. In the context of the ever-increasing urbanization of humanity, the densification of urban development and the growth of the number of storeys in buildings, the topic of the research seems extremely relevant. The authors focus on the development and testing of the pedestrian dead reckoning (PDR) method based on Transformer+TCN (Temporal Convolutional Network), as well as on factor graph optimization based on sliding windows. In general, the subject matter and level of research corresponds to the journal “Sensors”. But some designations and methods used in the study are not disclosed in sufficient detail. This needs to be corrected before the manuscript is published.
Comments:
1) The abstract does not disclose the abbreviations: Inertial Measurement Unit IMU and others (TCN and LSTM).
2) Figure 1. In Figure 1, the arrows from the sensors (Accelerometer, Gyroscope and Altimeter) overlap each other. This makes it difficult to understand the Framework diagram of PDR.
3) All notations used in expressions 1-13 should be explained in section 2.2. For example: 𝑣𝑝𝑑𝑟, 𝜃, 𝜏𝑔, 𝑅, 𝑞 and others.
3) Lines 186-187. The authors write “The main deterministic error terms of MIMU include zero bias, scale factor, threeaxis non-orthogonality error, etc.” This is incorrect. The scale factor is not an error. The error may be caused by nonlinearity or instability of the scale factor.
4) Section 2.3. You should indicate how the xyz coordinate system is oriented (Cartesian coordinate system, x-axis pointing north, etc.).
5) Section 2.5. The abbreviations MAG and ACC are not disclosed.
6) Section 3.1. Describe the test group in more detail. How many were men and women? Were there any disabled people among them? What was their weight, height, and age?
7) Line 424. The experiments were performed using an Intel Core i7-11800H and GPU: RTX 3060. That is, the results were not processed by the Huawei Mate30 Pro phone? If so, this should be clearly noted in the article, and the following questions should be discussed in sufficient detail: “Can the navigation methods proposed by the authors be implemented only using the Huawei Mate30 Pro phone or other modern phone models?”; “What are the minimum hardware requirements for the implementation of the described methods and algorithms?”
8) Beginning of section 4.1. Clearly and concisely describe how the measurement error values were determined. It should also be clear which sensor was used as a reference.
9) Figures 4 and 5. The figures contain marks in the form of red crosses. These marks should be explained in the text.
10) Figures 4 and 6. The axes of the figures must be labeled.
11) Figures 7 and 8. It is necessary to indicate (label) which parts of the figure show the results for “flat end”, “call”, “shake”, and “pocket”.
12) Figures 7, 9, 11 and 13. Incorrect designation of one of the axes (“vertical”). In the experiments, pedestrians move in a horizontal plane. Therefore, it is incorrect to use the term “vertical” to designate one of the axes of pedestrian movement.
13) For greater clarity of the study, the following questions should be discussed in the text of the manuscript: What accuracy of position estimation is sufficient for pedestrian navigation indoors? In what cases can the use of the methods proposed by the authors be justified for increasing (Absolute position error improves by only a few percent) the accuracy of pedestrian position estimation? Why is it not sufficient to use only an Ultra-wide Band sensor for pedestrian navigation in a building? How will the methods and algorithms proposed in the manuscript work taking into account the movement of pedestrians in elevators and on stairs? Is it planned to expand the test group in the future (a group of 10 people is too small for a confident statistical assessment of pedestrian navigation methods)?
Reviewer 3 Report
Comments and Suggestions for Authors
The paper presents valuable explorations in deep learning-based multi-sensor fusion for localization, demonstrating methodological novelty and providing essentially valid experimental verification. However, there remains room for improvement in theoretical depth, experimental comprehensiveness. While offering new technical insights for indoor navigation, additional experiments are required to validate its generalizability. My comments are as follows:
- The abstract contains abbreviations such as ACC, MAG that lack full definitions upon initial use. Standard academic practice requires spelling out abbreviations when first introduced.
- The contribution section fails to adequately highlight the novelty of the proposed algorithm, with overly generalized descriptions and insufficient technical details. Use experimental metrics (e.g., accuracy improvement percentage, computational efficiency) to demonstrate the method’s superiority.
- The derivations of Equations (1)-(12) require supplemental explanations of their physical meanings. Similar issues elsewhere in the text should be corrected accordingly.
- The experimental details are not clearly described, and data preprocessing is not mentioned.
- The authors should give the shortcomings of the proposed algorithm in the summary section.
Reviewer 4 Report
Comments and Suggestions for Authors
Overall, the paper provides a valuable contribution to indoor positioning research, but it requires some improvement through a better explanation of its novelty, a clearer review of related work, in additions to justification and explanation for the dataset used.
1- The literature review is incomplete and misses many recent papers on "Factor graph + IMU/Pedestrian Dead Reckoning", and "Factor Graph with PDR & Sliding Windows". Also, important prior works on step-length estimation using Transformer architectures are relevant but omitted. Besides, the review of machine learning approaches for step estimation remains generic; and a deeper comparison with techniques addressing cross-device or cross-user generalization is needed.
2- A clarification of novelty is crucial for the reader. The authors do not clearly differentiate their approach from existing hybrid approaches such as LSTM+Transformer or CNN+RNN frameworks used in recent studies.
3- The contribution of the sliding window–based factor graph optimization (FGO) is not well placed against prior FGO methods used for PDR.
4- The manuscript requires stating explicitly what aspects of the methodology introduced are new and how their improve upon previous work.
5- Aside from the introduction and literature review, certain aspects of the work may benefit from further specification to enhance applicability. Using a single smartphone model limits generalization across devices with different sensor behaviors. Data collection from ten individuals, without demographic context like age, gait, or physical attributes, may affect the model's robustness. And finally, the focus on four terminal modes (flat, call, handshake, pocket) omits transitional or dynamic scenarios (e.g., placing the device in a bag, jogging, stair navigation) common in real-world conditions. To more accurately position the contributions of this work, it would be helpful for the authors to acknowledge the noted limitations, contextualize them in relation to relevant prior studies, and consider framing them as directions for future research.
6- There is a typo in Figure 1: "Heading Angel" --> "Heading Angle"
7- Equations 1 through 13 appear technically valid, but their clarity could be enhanced as it may not be easily clear to all readers. For instance, Equation 6 requires an explicit definition of the skew-symmetric operator to ensure readers can fully see its role in the formulation. Similarly, the notation R(.) should be clearly described as representing the rotation matrix associated with a quaternion.
Round 2
Reviewer 1 Report
Comments and Suggestions for Authors
Regarding my comment 3), my understanding from the response is that the method is case-dependent, as it requires the training for individual cases to ensure the accuracy. This will make the proposed method not generic, but case-sensitive.
The authors do not address my comment 5), where I require the survey of advanced fusion techniques, based on which to justify the choose of the factor graph in the proposed method, as the factor graph is a kind of fusion techniques. However, the authors introduced advanced filtering rather than fusion techniques in the paper. For the clearness and authors' convenience, I provide the following representative advanced fusion techniques for the authors' reference: Distributed state fusion using sparse-grid quadrature filter with application to INS/CNS/GNSS integration, Multi-sensor optimal data fusion based on adaptive fading unscented Kalman filter, Multi-sensor optimal data fusion for INS/GNSS/CNS integration based on unscented Kalman filter, Matrix weighted multi-sensor data fusion for INS/GNSS/CNS integration, Modified federated Kalman filter for INS/GNSS/CNS integration, Random weighting method for multi-sensor data fusion, Random weighting estimation for fusion of multi-dimensional position data, Multi-sensor data fusion for INS/GPS/SAR integrated navigation system.
Author Response
Comments 5: Factor graph is a technique in the broad field of multi-sensor fusion. However, advanced multi-sensor fusion techniques are not introduced missing - e.g., advanced fusion techniques based on random weighting estimation, robust or adaptive UKF, and sparse-grid quadrature filter. It is recommended that the authors introduce advanced fusion techniques, based on which to justify the choice of factor graph in this paper. |
Response 5: In the Multi-sensor Fusion Methods section of the Introduction, I have added advanced fusion techniques such as the UKF, EKF, and MSCKF. I have also added a review of advanced fusion techniques, including: Distributed state fusion using sparse-grid quadrature filter with application to INS/CNS/GNSS integration, multi-sensor optimal data fusion based on adaptive fading unscented Kalman filter, Matrix weighted multi-sensor data fusion for INS/GNSS/CNS integration, and multi-sensor data fusion for INS/GPS/SAR integrated navigation system. line:115-132
|
Reviewer 4 Report
Comments and Suggestions for Authors
Upon reviewing the authors’ response and the revised manuscript, I confirm that the necessary modifications have been made. I have no further comments or suggestions.
Author Response
I have made some changes based on your comments. Thank you for your contribution.